# Promoting Coordination
# through Policy Regularization
# in Multi-Agent Deep Reinforcement Learning

**Julien Roy**[*]
Québec AI institute (Mila)
Polytechnique Montréal
`julien.roy@mila.quebec`

**Paul Barde**[*]
Québec AI institute (Mila)
McGill University
`bardepau@mila.quebec`

**Félix G. Harvey**
Québec AI institute (Mila)
Polytechnique Montréal

**Derek Nowrouzezahrai**
Québec AI institute (Mila)
McGill University

**Christopher Pal**[‡]
Québec AI institute (Mila)
Polytechnique Montréal
Element AI

## Abstract

In multi-agent reinforcement learning, discovering successful collective behaviors is challenging as it requires exploring a joint action space that grows exponentially with the number of agents. While the tractability of independent agent-wise exploration is appealing, this approach fails on tasks that require elaborate group strategies. We argue that coordinating the agents' policies can guide their exploration and we investigate techniques to promote such an inductive bias. We propose two policy regularization methods: TeamReg, which is based on inter-agent action predictability and CoachReg that relies on synchronized behavior selection. We evaluate each approach on four challenging continuous control tasks with sparse rewards that require varying levels of coordination as well as on the discrete action Google Research Football environment. Our experiments show improved performance across many cooperative multi-agent problems. Finally, we analyze the effects of our proposed methods on the policies that our agents learn and show that our methods successfully enforce the qualities that we propose as proxies for coordinated behaviors.

## 1   Introduction

Multi-Agent Reinforcement Learning (MARL) refers to the task of training an agent to maximize its expected return by interacting with an environment that contains other learning agents. It represents a challenging branch of Reinforcement Learning (RL) with interesting developments in recent years [11]. A popular framework for MARL is the use of a Centralized Training and a Decentralized Execution (CTDE) procedure [24, 8, 14, 7, 28]. Typically, one leverages centralized critics to approximate the value function of the aggregated observations-actions pairs and train actors restricted to the observation of a single agent. Such critics, if exposed to coordinated joint actions leading to high returns, can steer the agents' policies toward these highly rewarding behaviors. However, these approaches depend on the agents luckily stumbling on these collective actions in order to grasp their benefit. Thus, it might fail in scenarios where such behaviors are unlikely to occur by chance. We hypothesize that in such scenarios, coordination-promoting inductive biases on the policy search

---

[*]Equal contribution.
[‡]Canada CIFAR AI Chair.

could help discover successful behaviors more efficiently and supersede task-specific reward shaping and curriculum learning. To motivate this proposition we present a simple Markov Game in which agents forced to coordinate their actions learn remarkably faster. For more realistic tasks in which coordinated strategies cannot be easily engineered and must be learned, we propose to transpose this insight by relying on two coordination proxies to bias the policy search. The first avenue, TeamReg, assumes that an agent must be able to predict the behavior of its teammates in order to coordinate with them. The second, CoachReg, supposes that coordinated agents collectively recognize different situations and synchronously switch to different sub-policies to react to them.[2].

Our contributions are threefold. First, we show that coordination can crucially accelerate multi-agent learning for cooperative tasks. Second, we propose two novel approaches that aim at promoting such coordination by augmenting CTDE MARL algorithms through additional multi-agent objectives that act as policy regularizers and are optimized jointly with the main return-maximization objective. Third, we design two new sparse-reward cooperative tasks in the multi-agent particle environment [26]. We use them along with two standard multi-agent tasks to present a detailed evaluation of our approaches' benefits when they extend the reference CTDE MARL algorithm MADDPG [24]. We validate our methods' key components by performing an ablation study and a detailed analysis of their effect on agents' behaviors. Finally, we verify that these benefits hold on the more complex, discrete action, Google Research Football environment [20].

Our experiments suggest that our TeamReg objective provides a dense learning signal that can help guiding the policy towards coordination in the absence of external reward, eventually leading it to the discovery of higher performing team strategies in a number of cooperative tasks. However we also find that TeamReg does not lead to improvements in every single case and can even be harmful in environments with an adversarial component. For CoachReg, we find that enforcing synchronous sub-policy selection enables the agents to concurrently learn to react to different agreed upon situations and consistently yields significant improvements on the overall performance.

## 2 Background

### 2.1 Markov Games

We consider the framework of Markov Games [23], a multi-agent extension of Markov Decision Processes (MDPs). A Markov Game of $N$ agents is defined by the tuple $\langle \mathcal{S}, \mathcal{T}, \mathcal{P}, \{\mathcal{O}^i, \mathcal{A}^i, \mathcal{R}^i\}_{i=1}^N \rangle$ where $\mathcal{S}$, $\mathcal{T}$, and $\mathcal{P}$ are respectively the set of all possible states, the transition function and the initial state distribution. While these are global properties of the environment, $\mathcal{O}^i$, $\mathcal{A}^i$ and $\mathcal{R}^i$ are individually defined for each agent $i$. They are respectively the observation functions, the sets of all possible actions and the reward functions. At each time-step $t$, the global state of the environment is given by $s_t \in \mathcal{S}$ and every agent's individual action vector is denoted by $a_t^i \in \mathcal{A}^i$. To select their action, each agent $i$ only has access to its own observation vector $o_t^i$ which is extracted by the observation function $\mathcal{O}^i$ from the global state $s_t$. The initial state $s_0$ is sampled from the initial state distribution $\mathcal{P} : \mathcal{S} \to [0, 1]$ and the next state $s_{t+1}$ is sampled from the probability distribution over the possible next states given by the transition function $\mathcal{T} : \mathcal{S} \times \mathcal{S} \times \mathcal{A}^1 \times ... \times \mathcal{A}^N \to [0, 1]$. Finally, at each time-step, each agent receives an individual scalar reward $r_t^i$ from its reward function $\mathcal{R}^i : \mathcal{S} \times \mathcal{S} \times \mathcal{A}^1 \times ... \times \mathcal{A}^N \to \mathbb{R}$. Agents aim at maximizing their expected discounted return $\mathbb{E}\left[\sum_{t=0}^T \gamma^t r_t^i\right]$ over the time horizon $T$, where $\gamma \in [0, 1]$ is a discount factor.

### 2.2 Multi-Agent Deep Deterministic Policy Gradient

MADDPG [24] is an adaptation of the Deep Deterministic Policy Gradient algorithm [22] to the multi-agent setting. It allows the training of cooperating and competing decentralized policies through the use of a centralized training procedure. In this framework, each agent $i$ possesses its own deterministic policy $\mu^i$ for action selection and critic $Q^i$ for state-action value estimation, which are respectively parametrized by $\theta^i$ and $\phi^i$. All parametric models are trained off-policy from previous transitions $\zeta_t := (\mathbf{o}_t, \mathbf{a}_t, \mathbf{r}_t, \mathbf{o}_{t+1})$ uniformly sampled from a replay buffer $\mathcal{D}$. Note that $\mathbf{o}_t := [o_t^1, ..., o_t^N]$ is the joint observation vector and $\mathbf{a}_t := [a_t^1, ..., a_t^N]$ is the joint action vector, obtained by concatenating

the individual observation vectors $o_t^i$ and action vectors $a_t^i$ of all $N$ agents. Each centralized critic is trained to estimate the expected return for a particular agent $i$ from the Q-learning loss [33]:

$$\mathcal{L}^i(\phi^i) = \mathbb{E}_{\zeta_t \sim \mathcal{D}} \left[ \frac{1}{2} \left( Q^i(\mathbf{o}_t, \mathbf{a}_t; \phi^i) - y_t^i \right)^2 \right]$$

$$y_t^i = r_t^i + \gamma Q^i(\mathbf{o}_{t+1}, \mathbf{a}_{t+1}; \bar{\phi}^i) \Big|_{a_{t+1}^j = \mu^j(o_{t+1}^j; \bar{\theta}^j) \, \forall j} \tag{1}$$

For a given set of weights $w$, we define its target counterpart $\bar{w}$, updated from $\bar{w} \leftarrow \tau w + (1 - \tau)\bar{w}$ where $\tau$ is a hyper-parameter. Each policy is updated to maximize the expected discounted return of the corresponding agent $i$ :

$$J_{PG}^i(\theta^i) = \mathbb{E}_{\mathbf{o}_t \sim \mathcal{D}} \left[ Q^i(\mathbf{o}_t, \mathbf{a}_t) \Big|_{\substack{a_t^i = \mu^i(o_t^i; \theta^i), \\ a_t^j = \mu^j(o_t^j; \bar{\theta}^j) \, \forall j \neq i}} \right] \tag{2}$$

By taking into account *all* agents' observation-action pairs when guiding an agent's policy, the value-functions are trained in a centralized, stationary environment, despite taking place in a multi-agent setting. This mechanism can allow to learn coordinated strategies that can then be deployed in a decentralized way. However, this procedure does not encourage the *discovery* of coordinated strategies since high-return behaviors have to be randomly experienced through unguided exploration.

## 3 Motivation

In this section, we aim to answer the following question: can coordination help the discovery of effective policies in cooperative tasks? Intuitively, coordination can be defined as an agent's behavior being informed by the behavior of another agent, i.e. structure in the agents' interactions. Namely, a team where agents behave independently of one another would not be coordinated.

Consider the simple Markov Game consisting of a chain of length $L$ leading to a termination state as depicted in Figure 1. At each time-step, both agents receive $r_t = -1$. The joint action of these two agents in this environment is given by $\mathbf{a} \in \mathcal{A} = \mathcal{A}^1 \times \mathcal{A}^2$, where $\mathcal{A}^1 = \mathcal{A}^2 = \{0, 1\}$. Agent $i$ tries to go right when selecting $a^i = 0$ and left when selecting $a^i = 1$. However, to transition to a different state both agents need to perform the same action at the same time (two lefts or two rights). Now consider a slight variant of this environment with a different joint action structure $\mathbf{a}' \in \mathcal{A}'$. The former action structure is augmented with a hard-coded coordination module which maps the joint primitive $a^i$ to $a^{i\prime}$ like so:

$$\mathbf{a}' = \begin{pmatrix} a^{1\prime} = a^1 \\ a^{2\prime} = a^1 a^2 + (1 - a^1)(1 - a^2) \end{pmatrix}, \begin{pmatrix} a^1 \\ a^2 \end{pmatrix} \in \mathcal{A}$$

While the second agent still learns a state-action value function $Q^2(s, a^2)$ with $a^2 \in \mathcal{A}^2$, the coordination module builds $a^{2\prime}$ from $(a^1, a^2)$ so that $a^{2\prime}$ effectively determines whether the second agent acts in *agreement* or in *disagreement* with the first agent. In other words, if $a^2 = 1$, then $a^{2\prime} = a^1$ (agreement) and if $a^2 = 0$, then $a^{2\prime} = 1 - a^1$ (disagreement).

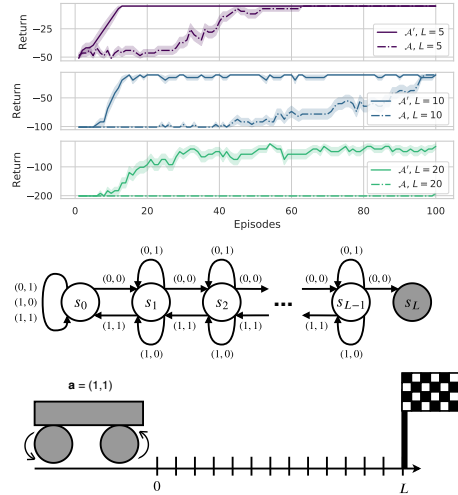

Figure 1: (Top) The tabular Q-learning agents learn much more efficiently when constrained to the space of coordinated policies (solid lines) than in the original action space (dashed lines). (Bottom) Simple Markov Game consisting of a chain of length $L$ leading to a terminal state (in grey). Agents can be seen as the two wheels of a vehicle so that their actions need to be in agreement for the vehicle to move. The detailed experimental setup is reported in Appendix A.

While it is true that this additional structure does not modify the action space nor the independence of the action selection, it reduces the stochasticity of the transition dynamics as seen by agent 2. In the first setup, the outcome of an agent's action is

conditioned on the action of the other agent. In the second setup, if agent 2 decides to disagree, the transition becomes deterministic as the outcome is independent of agent 1. This suggests that by reducing the entropy of the transition distribution, this mapping reduces the variance of the Q-updates and thus makes online tabular Q-learning agents learn much faster (Figure 1).

This example uses a handcrafted mapping in order to demonstrate the effectiveness of exploring in the space of coordinated policies rather than in the unconstrained policy space. Now, the following question remains: how can one softly learn the same type of constraint throughout training for any multi-agent cooperative tasks? In the following sections, we present two algorithms that tackle this problem.

# 4  Coordination and Policy regularization [3]

## 4.1  Team regularization

This first approach aims at exploiting the structure present in the joint action space of coordinated policies to attain a certain degree of predictability of one agent's behavior with respect to its teammate(s). It is based on the hypothesis that the reciprocal also holds i.e. that promoting agents' predictability could foster such team structure and lead to more coordinated behaviors. This assumption is cast into the decentralized framework by training agents to predict their teammates' actions given only their own observation. For continuous control, the loss is the mean squared error (MSE) between the predicted and true actions of the teammates, yielding a teammate-modelling secondary objective. For discrete action spaces, we use the KL-divergence ($D_{\text{KL}}$) between the predicted and real action distributions of an agent pair.

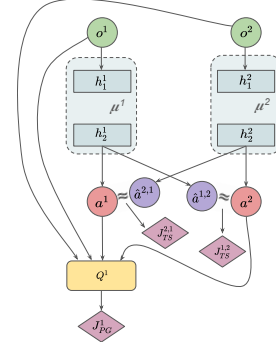

Figure 2: Illustration of TeamReg with two agents. Each agent's policy is equipped with additional heads that are trained to predict other agents' actions and every agent is regularized to produce actions that its teammates correctly predict. The method is depicted for agent 1 only to avoid cluttering.

While estimating teammates' policies can be used to enrich the learned representations, we extend this objective to also drive the teammates' behaviors towards the predictions by leveraging a differentiable action selection mechanism. We call *team-spirit* this objective pair $J_{TS}^{i,j}$ and $J_{TS}^{j,i}$ between agents $i$ and $j$:

$$J_{TS\text{-continuous}}^{i,j}(\theta^i, \theta^j) = -\mathbb{E}_{\mathbf{o}_t \sim \mathcal{D}}\left[\text{MSE}(\mu^j(o_t^j; \theta^j), \hat{\mu}^{i,j}(o_t^i; \theta^i))\right] \tag{3}$$

$$J_{TS\text{-discrete}}^{i,j}(\theta^i, \theta^j) = -\mathbb{E}_{\mathbf{o}_t \sim \mathcal{D}}\left[\text{D}_{\text{KL}}\left(\pi^j(\cdot|o_t^j; \theta^j)||\hat{\pi}^{i,j}(\cdot|o_t^i; \theta^i)\right)\right] \tag{4}$$

where $\hat{\mu}^{i,j}$ (or $\hat{\pi}^{i,j}$ in the discrete case) is the policy head of agent $i$ trying to predict the action of agent $j$. The total objective for a given agent $i$ becomes:

$$J_{total}^i(\theta^i) = J_{PG}^i(\theta^i) + \lambda_1 \sum_j J_{TS}^{i,j}(\theta^i, \theta^j) + \lambda_2 \sum_j J_{TS}^{j,i}(\theta^j, \theta^i) \tag{5}$$

where $\lambda_1$ and $\lambda_2$ are hyper-parameters that respectively weigh how well an agent should predict its teammates' actions, and how predictable an agent should be for its teammates. We call TeamReg this dual regularization from team-spirit objectives. Figure 2 summarizes these interactions.

## 4.2  Coach regularization

In order to foster coordinated interactions, this method aims at teaching the agents to recognize different situations and synchronously select corresponding sub-behaviors.

**Sub-policy selection** Firstly, to enable explicit sub-behavior selection, we propose the use of *policy masks* as a means to modulate the agents' policies. A policy mask $u^j$ is a one-hot vector of size $K$ (a fixed hyper-parameter) with its $j^{\text{th}}$ component set to one. In practice, we use these masks to perform dropout [30] in a structured manner on $\tilde{h}_1 \in \mathbb{R}^M$, the pre-activations of the first hidden layer $h_1$ of the policy network $\pi$. To do so, we construct the vector $\boldsymbol{u}^j$, which is the concatenation of $C$ copies of $u^j$, in order to reach the dimensionality $M = C * K$. The element-wise product $\boldsymbol{u}^j \odot \tilde{h}_1$ is performed and only the units of $\tilde{h}_1$ at indices $m$ modulo $K = j$ are kept for $m = 0, \ldots, M - 1$. Each agent $i$ generates $e_t^i$, its own policy mask from its observation $o_t^i$, to modulate its policy network. Here, a simple linear layer $l^i$ is used to produce a categorical probability distribution $p^i(e_t^i | o_t^i)$ from which the one-hot vector is sampled:

$$p^i(e_t^i = u^j | o_t^i) = \frac{\exp\left(l^i(o_t^i; \theta^i)_j\right)}{\sum_{k=0}^{K-1} \exp\left(l^i(o_t^i; \theta^i)_k\right)} \tag{6}$$

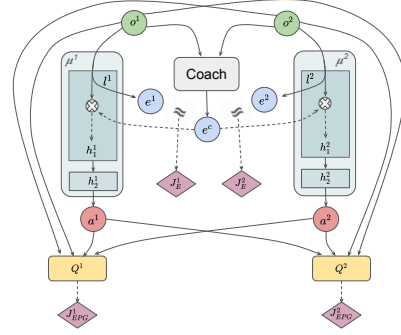

Figure 3: Illustration of CoachReg with two agents. A central model, the coach, takes all agents' observations as input and outputs the current mode (policy mask). Agents are regularized to predict the same mask from their local observations and optimize the corresponding sub-policy.

**Synchronous sub-policy selection** Although the policy masking mechanism enables the agent to swiftly switch between sub-policies it does not encourage the agents to synchronously modulate their behavior. To promote synchronicity we introduce the *coach* entity, parametrized by $\psi$, which learns to produce policy-masks $e_t^c$ from the joint observations, i.e. $p^c(e_t^c | \mathbf{o}_t; \psi)$. The coach is used at training time only and drives the agents toward synchronously selecting the same behavior mask. Specifically, the coach is trained to output masks that (1) yield high returns when used by the agents and (2) are predictable by the agents. Similarly, each agent is regularized so that (1) its private mask matches the coach's mask and (2) it derives efficient behavior when using the coach's mask. At evaluation time, the coach is removed and the agents only rely on their own policy masks. The policy gradient objective when agent $i$ is provided with the coach's mask is given by:

$$J_{EPG}^i(\theta^i, \psi) = \mathbb{E}_{\mathbf{o}_t, \mathbf{a}_t \sim \mathcal{D}} \left[ Q^i(\mathbf{o}_t, \mathbf{a}_t) \Big|_{\substack{a_t^i = \mu(o_t^i, e_t^c; \theta^i) \\ e_t^c \sim p^c(\cdot | \mathbf{o}_t; \psi)}} \right] \tag{7}$$

The difference between the mask distribution of agent $i$ and the coach's one is measured from the Kullback–Leibler divergence:

$$J_E^i(\theta^i, \psi) = -\mathbb{E}_{\mathbf{o}_t \sim \mathcal{D}} \left[ D_{\text{KL}}\left( p^c(\cdot | \mathbf{o}_t; \psi) \| p^i(\cdot | o_t^i; \theta^i) \right) \right] \tag{8}$$

The total objective for agent $i$ is:

$$J_{total}^i(\theta^i) = J_{PG}^i(\theta^i) + \lambda_1 J_E^i(\theta^i, \psi) + \lambda_2 J_{EPG}^i(\theta^i, \psi) \tag{9}$$

with $\lambda_1$ and $\lambda_2$ the regularization coefficients. Similarly, the coach is trained with the following dual objective, weighted by the $\lambda_3$ coefficient:

$$J_{total}^c(\psi) = \frac{1}{N} \sum_{i=1}^{N} \left( J_{EPG}^i(\theta^i, \psi) + \lambda_3 J_E^i(\theta^i, \psi) \right) \tag{10}$$

In order to propagate gradients through the sampled policy mask we reparameterized the categorical distribution using the Gumbel-softmax [15] with a temperature of 1. We call this coordinated sub-policy selection regularization CoachReg and illustrate it in Figure 3.

## 5 Related Work

Several works in MARL consider explicit communication channels between the agents and distinguish between communicative actions (e.g. broadcasting a given message) and physical actions (e.g. moving in a given direction) [6, 26, 21]. Consequently, they often focus on the emergence of

language, considering tasks where the agents must discover a common communication protocol to succeed. Deriving a successful communication protocol can already be seen as coordination in the communicative action space and can enable, to some extent, successful coordination in the physical action space [1]. Yet, explicit communication is not a necessary condition for coordination as agents can rely on physical communication [26, 9].

TeamReg falls in the line of work that explores how to shape agents' behaviors with respect to other agents through auxiliary tasks. Strouse et al. [31] use the mutual information between the agent's policy and a goal-independent policy to shape the agent's behavior towards hiding or spelling out its current goal. However, this approach is only applicable for tasks with an explicit goal representation and is not specifically intended for coordination. Jaques et al. [16] approximate the direct causal effect between agent's actions and use it as an intrinsic reward to encourage social empowerment. This approximation relies on each agent learning a model of other agents' policies to predict its effect on them. In general, this type of behavior prediction can be referred to as *agent modelling* (or opponent modelling) and has been used in previous work to enrich representations [12, 13], to stabilise the learning dynamics [10] or to classify the opponent's play style [29].

With CoachReg, agents learn to unitedly recognize different modes in the environment and adapt by jointly switching their policy. This echoes with the hierarchical RL litterature and in particular with the single agent options framework [3] where the agent switches between different sub-policies, the options, depending on the current state. To encourage cooperation in the multi-agent setting, Ahilan and Dayan [1] proposed that an agent, the "manager", is extended with the possibility of setting other agents' rewards in order to guide collaboration. CoachReg stems from a similar idea: reaching a consensus is easier with a central entity that can asymmetrically influence the group. Yet, Ahilan and Dayan [1] guides the group in terms of "ends" (influences through the rewards) whereas CoachReg constrains it in terms of "means" (the group must synchronously switch between different strategies). Hence, the interest of CoachReg does not just lie in training sub-policies (which are obtained here through a simple and novel masking procedure) but rather in co-evolving synchronized sub-policies across multiple agents. Mahajan et al. [25] also looks at sub-policies co-evolution to tackle the problem of joint exploration, however their selection mechanism occurs only on the first timestep and requires duplicating random seeds across agents at test time. On the other hand, with CoachReg the sub-policy selection is explicitly decided by the agents themselves at each timestep without requiring a common sampling procedure since the mode recognition has been learned and grounded on the state throughout training.

Finally, Barton et al. [4] propose convergent cross mapping (CCM) to measure the degree of effective coordination between two agents. Although this represents an interesting avenue for behavior analysis, it fails to provide a tool for effectively enforcing coordination as CCM must be computed over long time series making it an impractical learning signal for single-step temporal difference methods.

To our knowledge, this work is the first to extend agent modelling to derive an inductive bias towards team-predictable policies or to introduce a collective, agent induced, modulation of the policies without an explicit communication channel. Importantly, these coordination proxies are enforced throughout training only, which allows to maintain decentralised execution at test time.

# 6 Training environments

Our continuous control tasks are built on OpenAI's multi-agent particle environment [26]. SPREAD and CHASE were introduced by [24]. We use SPREAD as is but with sparse rewards. CHASE is modified with a prey controlled by repulsion forces so that only the predators are learnable, as we wish to focus on coordination in cooperative tasks. Finally we introduce COMPROMISE and BOUNCE where agents are physically tied together. While positive return can be achieved in these tasks by selfish agents, they all benefit from coordinated strategies and maximal return can only be achieved by agents working closely together. Figure 4 presents a visualization and a brief description. In all tasks, agents receive as observation their own global position and velocity as well as the relative position of other entities. A more detailed description is provided in Appendix B. Note that work showcasing experiments on these environments often use discrete action spaces and dense rewards (e.g. the proximity with the objective) [14, 24, 17]. In our experiments, agents learn with continuous action spaces and from sparse rewards which is a far more challenging setting.

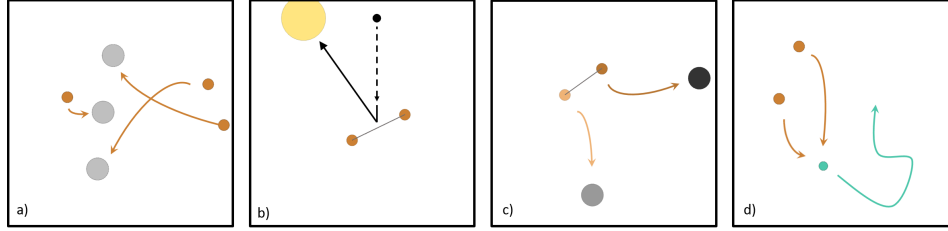

Figure 4: Multi-agent tasks we employ. (a) SPREAD: Agents must spread out and cover a set of landmarks. (b) BOUNCE: Two agents are linked together by a spring and must position themselves so that the falling black ball bounces towards a target. (c) COMPROMISE: Two linked agents must compete or cooperate to reach their own assigned landmark. (d) CHASE: Two agents chase a (non-learning) prey (turquoise) that moves w.r.t repulsion forces from predators and walls.

## 7    Results and Discussion

The proposed methods offer a way to incorporate new inductive biases in CTDE multi-agent policy search algorithms. We evaluate them by extending MADDPG, one of the most widely used algorithm in the MARL literature. We compare against vanilla MADDPG as well as two of its variants in the four cooperative multi-agent tasks described in Section 6. The first variant (DDPG) is the single-agent counterpart of MADDPG (decentralized training). The second (MADDPG + sharing) shares the policy and value-function models across agents. Additionally to the two proposed algorithms and the three baselines, we present results for two ablated versions of our methods. The first ablation (MADDPG + agent modelling) is similar to TeamReg but with $\lambda_2 = 0$, which results in only enforcing agent modelling and not encouraging agent predictability. The second ablation (MADDPG + policy mask) uses the same policy architecture as CoachReg, but with $\lambda_{1,2,3} = 0$, which means that agents still predict and apply a mask to their own policy, but synchronicity is not encouraged.

To offer a fair comparison between all methods, the hyper-parameter search routine is the same for each algorithm and environment (see Appendix E.1). For each search-experiment (one per algorithm per environment), 50 randomly sampled hyper-parameter configurations each using 3 random seeds are used to train the models for $15,000$ episodes. For each algorithm-environment pair, we then select the best hyper-parameter configuration for the final comparison and retrain them on 10 seeds for twice as long. This thorough evaluation procedure represents around 3 CPU-year. We give all details about the training setup and model selection in Appendix C and E.2. The results of the hyper-parameter searches are given in Appendix E.5. Interestingly, Figure 9 shows that our proposed coordination regularizers improve robustness to hyper-parameters despite having more hyper-parameters to tune.

### 7.1    Asymptotic Performance

Figure 5 reports the average learning curves and Table 1 presents the final performance. CoachReg is the best performing algorithm considering performance across all tasks. TeamReg also significantly improves performance on two tasks (SPREAD and BOUNCE) but shows unstable behavior on COMPROMISE, the only task with an adversarial component. This result reveals one limitation of this approach and is dicussed in details in Appendix F. Note that all algorithms perform similarly well on CHASE, with a slight advantage to the one using parameter sharing; yet this superiority is restricted to this task where the optimal strategy is to move symmetrically and squeeze the prey into a corner. Contrary to popular belief, we find that MADDPG almost never significantly outperforms DDPG in these sparse reward environments, supporting the hypothesis that while CTDE algorithms can in principle identify and reinforce highly rewarding coordinated behavior, they are likely to fail to do so if not incentivized to coordinate.

Regarding the ablated versions of our methods, the use of unsynchronized policy masks might result in swift and unpredictable behavioral changes and make it difficult for agents to perform together and coordinate. Experimentally, "MADDPG + policy mask" performs similarly or worse than MADDPG on all but one environment, and never outperforms the full CoachReg approach. However, policy masks alone seem sufficient to succeed on SPREAD, which is about selecting a landmark from a set. Finally "MADDPG + agent modelling" does not drastically improve on MADDPG apart from

Table 1: Final performance reported as mean return over agents averaged across 10 episodes and 10 seeds (± SE).

| env \ alg | DDPG | MADDPG | MADDPG +sharing | MADDPG +agent modelling | MADDPG +policy mask | MADDPG +TeamReg (ours) | MADDPG +CoachReg (ours) |
|---|---|---|---|---|---|---|---|
| SPREAD | $133 \pm 12$ | $159 \pm 6$ | $47 \pm 8$ | $183 \pm 10$ | $\mathbf{221 \pm 11}$ | $216 \pm 12$ | $210 \pm 12$ |
| BOUNCE | $3.6 \pm 1.4$ | $4.0 \pm 1.6$ | $0.0 \pm 0.0$ | $3.8 \pm 1.5$ | $3.7 \pm 1.1$ | $\mathbf{5.8 \pm 1.3}$ | $7.4 \pm 1.2$ |
| COMPROMISE | $19.1 \pm 1.2$ | $18.1 \pm 1.1$ | $19.6 \pm 1.5$ | $12.9 \pm 0.9$ | $18.4 \pm 1.3$ | $8.8 \pm 0.9$ | $\mathbf{31.1 \pm 1.1}$ |
| CHASE | $727 \pm 87$ | $834 \pm 80$ | $\mathbf{980 \pm 64}$ | $946 \pm 69$ | $722 \pm 82$ | $917 \pm 90$ | $949 \pm 54$ |

one environment, and is always outperformed by the full TeamReg (except on COMPROMISE, see Appendix F) which supports the importance of enforcing predictability alongside agent modeling.

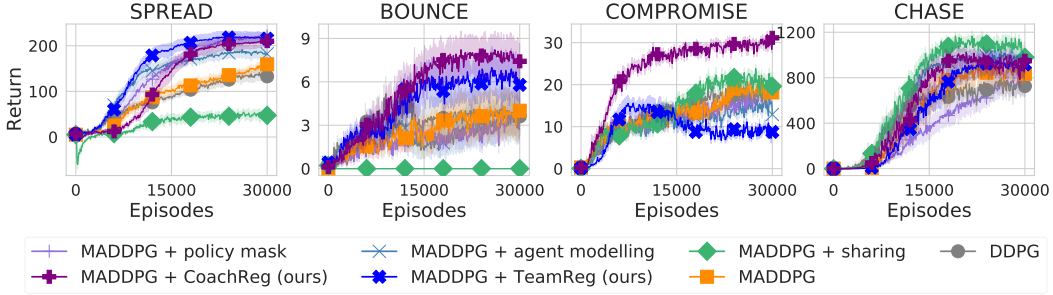

Figure 5: Learning curves (mean return over agents) for our two proposed algorithms, two ablations and three baselines on all four environments. Solid lines are the mean and envelopes are the Standard Error (SE) across the 10 training seeds.

## 7.2 Effects of enforcing predictable behavior

Here we validate that enforcing predictability makes the agent-modelling task more successful. To this end, we compare, on the SPREAD environment, the team-spirit losses between TeamReg and its ablated versions. Figure 6 shows that initially, due to the weight initialization, the predicted and actual actions both have relatively small norms yielding small values of team-spirit loss. As training goes on ($\sim$1000 episodes), the norms of the action-vector increase and the regularization loss becomes more important. As expected, MADDPG leads to the worst team-spirit loss as it is not trained to predict the actions of other agents. When using only the agent-modelling objective ($\lambda_1 > 0$), the agents significantly decrease the team-spirit loss, but it never reaches values as low as when using the full TeamReg objective ($\lambda_1 > 0$ and $\lambda_2 > 0$). Note that the team-spirit loss increases when performance starts to improve i.e. when agents start to master the task ($\sim$8000 episodes). Indeed, once the return maximisation signal becomes stronger, the relative importance of the auxiliary objective is reduced. Being predictable with respect to one-another may push agents to explore in a more structured and informed manner in the absence of reward signal, as similarly pursued by intrinsic motivation approaches [5].

## 7.3 Analysis of synchronous sub-policy selection

In this section we confirm that CoachReg yields the desired behavior: agents *synchronously* alternating between *varied* sub-policies.

Figure 7 shows the average entropy of the mask distributions for each environment compared to the entropy of Categorical Uniform Distributions of size $k$ ($k$-CUD). On all the environments, agents use several masks and tend to alternate between masks with more variety (close to uniformly switching between 3 masks) on SPREAD (where there are 3 agents and 3 goals) than on the other environments (comprised of 2 agents). Moreover, the Hamming proximity between the agents' mask sequences, $1 - D_h$ where $D_h$ is the Hamming distance (i.e. the ratio of timesteps for which the two sequences are different) shows that agents are synchronously selecting the same policy mask at test time (without a coach). Finally, we observe that some settings result in the agents coming up with interpretable strategies, like the one depicted in Figure 13 in Appendix G.2 where the agents alternate between two sub-policies depending on the position of the target[4].

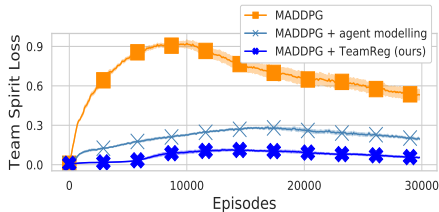

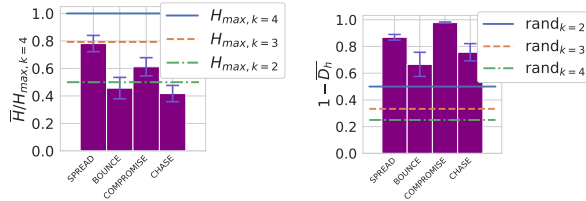

Figure 6: Effect of enabling and disabling the coefficients $\lambda_1$ and $\lambda_2$ on the ability of agents to predict their teammates behavior. Solid lines and envelope are average and SE on 10 seeds on SPREAD.

Figure 7: (Left) Average entropy of the policy mask distributions for each task. $H_{max,k}$ is the entropy of a $k$-CUD. (Right) Average Hamming Proximity between the policy mask sequence of agent pairs. rand$_k$ stands for agents independently sampling their masks from $k$-CUD. Error bars are the SE on 10 seeds.

### 7.4 Experiments on discrete action spaces

We evaluate our techniques on the more challenging task of 3vs2 Google Research football environment [20]. In this environment, each agent controls an offensive player and tries to score against a defensive player and a goalkeeper controlled by the engine's rule-based bots. Here agents have discrete action spaces of size 21, with actions like moving direction, dribble, sprint, short pass, high pass, etc. We use as observations 37-dimensional vectors containing players' and ball's coordinates, directions, etc.

The algorithms presented in Table 2 were trained using 25 randomly sampled hyperparameter configurations. The best configuration was retrained using 10 seeds for 80,000 episodes of 100 steps. Table 2 shows the mean return ($\pm$ standard error across seeds) on the last 10,000 episodes. All algorithms but MADDPG + CoachReg fail to reliably learn policies that achieve positive return (i.e. scoring goals).

Table 2: Average Returns for 3v2 football

| MADDPG | $0.004 \pm 0.002$ |
|---|---|
| MADDPG + sharing | $0.005 \pm 0.003$ |
| MADDPG + TeamReg (ours) | $0.006 \pm 0.003$ |
| MADDPG + CoachReg (ours) | $\mathbf{0.088 \pm 0.017}$ |

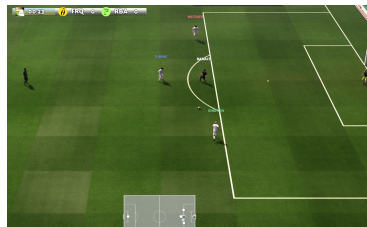

Figure 8: Snapshot of the google research football *3vs1-with-keeper*.

## 8 Conclusion

In this work we motivate the use of coordinated policies to ease the discovery of successful strategies in cooperative multi-agent tasks and propose two distinct approaches to promote coordination for CTDE multi-agent RL algorithms. While the benefits of TeamReg appear task-dependent – we show for example that it can be detrimental on tasks with a competitive component – CoachReg significantly improves performance on almost all presented environments. Motivated by the success of this single-step coordination technique, a promising direction is to explore model-based planning approaches to promote coordination over long-term multi-agent interactions.

## Broader Impact

In this work, we present and study methods to enforce coordination in MARL algorithms. It goes without saying that multi-agent systems can be employed for positive and negative applications alike. We do not propose methods aimed at making new applications possible or improving a particular set of applications. We instead propose methods that allow to better understand and improve multi-agent RL algorithms in general. Therefore, we do not aim in this section at discussing the impact of Multi-Agent Reinforcement Learning applications themselves but focus on the impact of our contribution: promoting multi-agent behaviors that are coordinated.

We first observe that current Multi-Agent Reinforcement Learning (MARL) algorithms may fail to train agents that leverage information about the behavior of their teammates and that even when

explicitly given their teammates observations, action and current policy during the training phase. We believe that this is an important observation worth raising some concern among the community since there is a widespread belief that centralized training (like MADDPG) should always outperform decentralize training (DDPG). Not only is this belief unsupported by empirical evidence (at least in our experiments) but it also prevents the community from investigating and tackling this flaw that is an important limitation for learning safer and more effective multi-agent behavior. By not accounting for the behavior of its teammates, an agent could not adapt to a new teammate or even a change in the teammates behavior. This prevents current methods to be applied in the real world where there is external perturbations and uncertainties and where an artificial agent may need to interact with various different individuals.

We propose to focus on coordination and sketch a definition of coordination: an agent behavior should be predictable given its teammate behavior. While this definition is restrictive, we believe that it is a good starting point to consider. Indeed, enforcing that criterion should make learning agents more aware of their teammates in order to coordinate with them. Yet, coordination alone does not ensure success, as agents could be coordinated in an unproductive manner. More so, coordination could have detrimental effects if it enables an attacker to influence an agent through taking control of a teammate or using a mock-up teammate. For these reasons, when using multi-agent RL algorithms (or even single-agent RL for that matter) for real world applications, additional safeguards are absolutely required to prevent the system from misbehaving, which is highly probable if out-of-distribution states are to be encountered.

## Acknowledgments and Disclosure of Funding

We thank Olivier Delalleau for his insightful feedback and comments. We also acknowledge funding in support of this work from the Fonds de Recherche Nature et Technologies (FRQNT) and Mitacs, as well as Compute Canada for supplying computing resources.

## Footnotes

[2]Source code for the algorithms and environments will be made public upon publication of this work. Visualisations of CoachReg are available here: `https://sites.google.com/view/marl-coordination/`

[3]Pseudocodes of our implementations are provided in Appendix D (see Algorithms 1 and 2).

[4]See animations at `https://sites.google.com/view/marl-coordination/`

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
