[Supplementary Material]

# A Additional details for experiment presented in Section 3 Motivation

We trained each agent $i$ with online Q-learning [33] on the $Q^i(a^i, s)$ table using Boltzmann exploration [18]. The Boltzmann temperature is fixed to 1 and we set the learning rate to 0.05 and the discount factor to 0.99. After each learning episode we evaluate the current greedy policy on 10 episodes and report the mean return. Curves are averaged over 20 seeds and the shaded area represents the standard error.

# B Tasks descriptions

**SPREAD** (Figure 4a): In this environment, there are 3 agents (small orange circles) and 3 landmarks (bigger gray circles). At every timestep, agents receive a team-reward $r_t = n - c$ where $n$ is the number of landmarks occupied by at least one agent and $c$ the number of collisions occurring at that timestep. To maximize their return, agents must therefore spread out and cover all landmarks. Initial agents' and landmarks' positions are random. Termination is triggered when the maximum number of timesteps is reached.

**BOUNCE** (Figure 4b): In this environment, two agents (small orange circles) are linked together with a spring that pulls them toward each other when stretched above its relaxation length. At episode's mid-time a ball (smaller black circle) falls from the top of the environment. Agents must position correctly so as to have the ball bounce on the spring towards the target (bigger beige circle), which turns yellow if the ball's bouncing trajectory passes through it. They receive a team-reward of $r_t = 0.1$ if the ball reflects towards the side walls, $r_t = 0.2$ if the ball reflects towards the top of the environment, and $r_t = 10$ if the ball reflects towards the target. At initialisation, the target's and ball's vertical position is fixed, their horizontal positions are random. Agents' initial positions are also random. Termination is triggered when the ball is bounced by the agents or when the maximum number of timesteps is reached.

**COMPROMISE** (Figure 4c): In this environment, two agents (small orange circles) are linked together with a spring that pulls them toward each other when stretched above its relaxation length. They both have a distinct assigned landmark (light gray circle for light orange agent, dark gray circle for dark orange agent), and receive a reward of $r_t = 10$ when they reach it. Once a landmark is reached by its corresponding agent, the landmark is randomly relocated in the environment. Initial positions of agents and landmark are random. Termination is triggered when the maximum number of timesteps is reached.

**CHASE** (Figure 4d): In this environment, two predators (orange circles) are chasing a prey (turquoise circle). The prey moves with respect to a scripted policy consisting of repulsion forces from the walls and predators. At each timestep, the learning agents (predators) receive a team-reward of $r_t = n$ where $n$ is the number of predators touching the prey. The prey has a greater max speed and acceleration than the predators. Therefore, to maximize their return, the two agents must coordinate in order to squeeze the prey into a corner or a wall and effectively trap it there. Termination is triggered when the maximum number of time steps is reached.

# C Training details

In all of our experiments, we use the Adam optimizer [19] to perform parameter updates. All models (actors, critics and coach) are parametrized by feedforward networks containing two hidden layers of 128 units. We use the Rectified Linear Unit (ReLU) [27] as activation function and layer normalization [2] on the pre-activations unit to stabilize the learning. We use a buffer-size of $10^6$ entries and a batch-size of $1024$. We collect 100 transitions by interacting with the environment for each learning update. For all tasks in our hyper-parameter searches, we train the agents for $15,000$ episodes of 100 steps and then re-train the best configuration for each algorithm-environment pair for twice as long ($30,000$ episodes) to ensure full convergence for the final evaluation. The scale of the exploration noise is kept constant for the first half of the training time and then decreases linearly to $0$ until the end of training. We use a discount factor $\gamma$ of 0.95 and a gradient clipping threshold of 0.5 in all experiments. Finally for CoachReg, we fixed $K$ to 4 meaning that agents could choose between 4 sub-policies. Since policies' hidden layers are of size 128 the corresponding value for $C$ is 32. All experiments were run on Intel E5-2683 v4 Broadwell (2.1GHz) CPUs in less than 12 hours.

# D Algorithms

---

**Algorithm 1** Team

---

Randomly initialize $N$ critic networks $Q^i$ and actor networks $\mu^i$
Initialize the target weights
Initialize one replay buffer $\mathcal{D}$
**for** episode from 0 to number of episodes **do**
    Initialize random processes $\mathcal{N}^i$ for action exploration
    Receive initial joint observation $\mathbf{o}_0$
    **for** timestep t from 0 to episode length **do**
        Select action $a_i = \mu^i(o_t^i) + \mathcal{N}_t^i$ for each agent
        Execute joint action $\mathbf{a}_t$ and observe joint reward $\mathbf{r}_t$ and new observation $\mathbf{o}_{t+1}$
        Store transition $(\mathbf{o}_t, \mathbf{a}_t, \mathbf{r}_t, \mathbf{o}_{t+1})$ in $\mathcal{D}$
    **end for**
    Sample a random minibatch of $M$ transitions from $\mathcal{D}$
    **for** each agent $i$ **do**
        Evaluate $\mathcal{L}^i$ and $J_{PG}^i$ from Equations (1) and (2)
        **for** each other agent $(j \neq i)$ **do**
            Evaluate $J_{TS}^{i,j}$ from Equations (3, 4)
            Update actor $j$ with $\theta^j \leftarrow \theta^j + \alpha_\theta \nabla_{\theta^j} \lambda_2 J_{TS}^{i,j}$
        **end for**
        Update critic with $\phi^i \leftarrow \phi^i - \alpha_\phi \nabla_{\phi^i} \mathcal{L}^i$
        Update actor $i$ with $\theta^i \leftarrow \theta^i + \alpha_\theta \nabla_{\theta^i} \left( J_{PG}^i + \lambda_1 \sum_{j=1}^N J_{TS}^{i,j} \right)$
    **end for**
    Update all target weights
**end for**

---

---

**Algorithm 2** Coach

---

Randomly initialize $N$ critic networks $Q^i$, actor networks $\mu^i$ and one coach network $p^c$
Initialize $N$ target networks $Q^{i\prime}$ and $\mu^{i\prime}$
Initialize one replay buffer $\mathcal{D}$
**for** episode from 0 to number of episodes **do**
    Initialize random processes $\mathcal{N}^i$ for action exploration
    Receive initial joint observation $\mathbf{o}_0$
    **for** timestep t from 0 to episode length **do**
        Select action $a_i = \mu^i(o_t^i) + \mathcal{N}_t^i$ for each agent
        Execute joint action $\mathbf{a}_t$ and observe joint reward $\mathbf{r}_t$ and new observation $\mathbf{o}_{t+1}$
        Store transition $(\mathbf{o}_t, \mathbf{a}_t, \mathbf{r}_t, \mathbf{o}_{t+1})$ in $\mathcal{D}$
    **end for**
    Sample a random minibatch of $M$ transitions from $\mathcal{D}$
    **for** each agent $i$ **do**
        Evaluate $\mathcal{L}^i$ and $J_{PG}^i$ from Equations (1) and (2)
        Update critic with $\phi^i \leftarrow \phi^i - \alpha_\phi \nabla_{\phi^i} \mathcal{L}^i$
        Update actor with $\theta^i \leftarrow \theta^i + \alpha_\theta \nabla_{\theta^i} J_{PG}^i$
    **end for**
    **for** each agent $i$ **do**
        Evaluate $J_E^i$ and $J_{EPG}^i$ from Equations (8) and (7)
        Update actor with $\theta^i \leftarrow \theta^i + \alpha_\theta \nabla_{\theta^i} \left( \lambda_1 J_E^i + \lambda_2 J_{EPG}^i \right)$
    **end for**
    Update coach with $\psi \leftarrow \psi + \alpha_\psi \nabla_\psi \frac{1}{N} \sum_{i=1}^N \left( J_{EPG}^i + \lambda_3 J_E^i \right)$
    Update all target weights
**end for**

---

# E Hyper-parameter search

## E.1 Hyper-parameter search ranges

We perform searches over the following hyper-parameters: the learning rate of the actor $\alpha_\theta$, the learning rate of the critic $\omega_\phi$ relative to the actor ($\alpha_\phi = \omega_\phi * \alpha_\theta$), the target-network soft-update parameter $\tau$ and the initial scale of the exploration noise $\eta_{noise}$ for the Ornstein-Uhlenbeck noise generating process [32] as used by Lillicrap et al. [22]. When using TeamReg and CoachReg, we additionally search over the regularization weights $\lambda_1$, $\lambda_2$ and $\lambda_3$. The learning rate of the coach is always equal to the actor's learning rate (i.e. $\alpha_\theta = \alpha_\psi$), motivated by their similar architectures and learning signals and in order to reduce the search space. Table 2 shows the ranges from which values for the hyper-parameters are drawn uniformly during the searches.

Table 2: Ranges for hyper-parameter search, the log base is 10

| HYPER-PARAMETER | RANGE |
|---|---|
| $\log(\alpha_\theta)$ | $[-8, -3]$ |
| $\log(\omega_\phi)$ | $[-2,\ \ 2]$ |
| $\log(\tau)$ | $[-3, -1]$ |
| $\log(\lambda_1)$ | $[-3,\ \ 0]$ |
| $\log(\lambda_2)$ | $[-3,\ \ 0]$ |
| $\log(\lambda_3)$ | $[-1,\ \ 1]$ |
| $\eta_{noise}$ | $[0.3, 1.8]$ |

## E.2 Model selection

During training, a policy is evaluated on a set of 10 different episodes every 100 learning steps. At the end of the training, the model at the best evaluation iteration is saved as the best version of the policy for this training, and is re-evaluated on 100 different episodes to have a better assessment of its final performance. The performance of a hyper-parameter configuration is defined as the average performance (across seeds) of the best policies learned using this set of hyper-parameter values.

### E.3 Selected hyper-parameters

Tables 3, 4, 5, and 6 shows the best hyper-parameters found by the random searches for each of the environments and each of the algorithms.

Table 3: Best found hyper-parameters for the SPREAD environment

| Hyper-parameter | DDPG | MADDPG | MADDPG+Sharing | MADDPG+TeamReg | MADDPG+CoachReg |
|---|---|---|---|---|---|
| $\alpha_\theta$ | $5.3 * 10^{-5}$ | $2.1 * 10^{-5}$ | $9.0 * 10^{-4}$ | $2.5 * 10^{-5}$ | $1.2 * 10^{-5}$ |
| $\omega_\phi$ | 53 | 79 | 0.71 | 42 | 82 |
| $\tau$ | 0.05 | 0.083 | 0.076 | 0.098 | 0.0077 |
| $\lambda_1$ | - | - | - | 0.054 | 0.13 |
| $\lambda_2$ | - | - | - | 0.29 | 0.24 |
| $\lambda_3$ | - | - | - | - | 8.4 |
| $\eta_{noise}$ | 1.0 | 0.5 | 0.7 | 1.2 | 1.6 |

Table 4: Best found hyper-parameters for the BOUNCE environment

| Hyper-parameter | DDPG | MADDPG | MADDPG+Sharing | MADDPG+TeamReg | MADDPG+CoachReg |
|---|---|---|---|---|---|
| $\alpha_\theta$ | $8.1 * 10^{-4}$ | $3.8 * 10^{-5}$ | $1.2 * 10^{-4}$ | $1.3 * 10^{-5}$ | $6.8 * 10^{-5}$ |
| $\omega_\phi$ | 2.4 | 87 | 0.47 | 85 | 9.4 |
| $\tau$ | 0.089 | 0.016 | 0.06 | 0.055 | 0.02 |
| $\lambda_1$ | - | - | - | 0.06 | 0.0066 |
| $\lambda_2$ | - | - | - | 0.0026 | 0.23 |
| $\lambda_3$ | - | - | - | - | 0.34 |
| $\eta_{noise}$ | 1.2 | 0.9 | 1.2 | 1.0 | 1.1 |

Table 5: Best found hyper-parameters for the CHASE environment

| Hyper-parameter | DDPG | MADDPG | MADDPG+Sharing | MADDPG+TeamReg | MADDPG+CoachReg |
|---|---|---|---|---|---|
| $\alpha_\theta$ | $4.5 * 10^{-4}$ | $2.0 * 10^{-4}$ | $9.7 * 10^{-4}$ | $1.3 * 10^{-5}$ | $1.8 * 10^{-4}$ |
| $\omega_\phi$ | 32 | 64 | 0.79 | 85 | 90 |
| $\tau$ | 0.031 | 0.021 | 0.032 | 0.055 | 0.011 |
| $\lambda_1$ | - | - | - | 0.06 | 0.0069 |
| $\lambda_2$ | - | - | - | 0.0026 | 0.86 |
| $\lambda_3$ | - | - | - | - | 0.76 |
| $\eta_{noise}$ | 0.6 | 1.0 | 1.5 | 1.0 | 1.1 |

Table 6: Best found hyper-parameters for the COMPROMISE environment

| Hyper-parameter | DDPG | MADDPG | MADDPG+Sharing | MADDPG+TeamReg | MADDPG+CoachReg |
|---|---|---|---|---|---|
| $\alpha_\theta$ | $6.1 * 10^{-5}$ | $3.1 * 10^{-4}$ | $6.2 * 10^{-4}$ | $1.5 * 10^{-5}$ | $3.4 * 10^{-4}$ |
| $\omega_\phi$ | 1.7 | 0.94 | 0.58 | 90 | 29 |
| $\tau$ | 0.065 | 0.045 | 0.007 | 0.02 | 0.0037 |
| $\lambda_1$ | - | - | - | 0.0013 | 0.65 |
| $\lambda_2$ | - | - | - | 0.56 | 0.5 |
| $\lambda_3$ | - | - | - | - | 1.3 |
| $\eta_{noise}$ | 1.1 | 0.7 | 1.3 | 1.6 | 1.6 |

Table 7: Best found hyper-parameters for the *3-vs-1-with-keeper* Google Football environment

| Hyper-parameter | MADDPG | MADDPG+Sharing | MADDPG+TeamReg | MADDPG+CoachReg |
|---|---|---|---|---|
| $\alpha_\theta$ | $1.6 * 10^{-6}$ | $3.4 * 10^{-5}$ | $3.5 * 10^{-6}$ | $9.4 * 10^{-5}$ |
| $\omega_\phi$ | 3.1 | 13 | 0.96 | 2.9 |
| $\tau$ | 0.004 | 0.0014 | 0.0066 | 0.018 |
| $\lambda_1$ | - | - | 0.1 | 0.027 |
| $\lambda_2$ | - | - | 0.02 | 0.027 |
| $\lambda_3$ | - | - | - | 2.4 |

### E.4 Selected hyper-parameters (ablations)

Tables 8, 9, 10, and 11 shows the best hyper-parameters found by the random searches for each of the environments and each of the ablated algorithms.

Table 8: Best found hyper-parameters for the SPREAD environment

| HYPER-PARAMETER | MADDPG+AGENT MODELLING | MADDPG+POLICY MASK |
| --- | --- | --- |
| $\alpha_\theta$ | $1.3 * 10^{-5}$ | $6.8 * 10^{-5}$ |
| $\omega_\phi$ | 85 | 9.4 |
| $\tau$ | 0.055 | 0.02 |
| $\lambda_1$ | 0.06 | 0 |
| $\lambda_2$ | 0 | 0 |
| $\lambda_3$ | - | 0 |
| $\eta_{noise}$ | 1.0 | 1.1 |

Table 9: Best found hyper-parameters for the BOUNCE environment

| HYPER-PARAMETER | MADDPG+AGENT MODELLING | MADDPG+POLICY MASK |
| --- | --- | --- |
| $\alpha_\theta$ | $1.3 * 10^{-5}$ | $2.5 * 10^{-4}$ |
| $\omega_\phi$ | 85 | 0.52 |
| $\tau$ | 0.055 | 0.0077 |
| $\lambda_1$ | 0.06 | 0 |
| $\lambda_2$ | 0 | 0 |
| $\lambda_3$ | - | 0 |
| $\eta_{noise}$ | 1.0 | 1.3 |

Table 10: Best found hyper-parameters for the CHASE environment

| HYPER-PARAMETER | MADDPG+AGENT MODELLING | MADDPG+POLICY MASK |
| --- | --- | --- |
| $\alpha_\theta$ | $2.5 * 10^{-5}$ | $6.8 * 10^{-5}$ |
| $\omega_\phi$ | 42 | 9.4 |
| $\tau$ | 0.098 | 0.02 |
| $\lambda_1$ | 0.054 | 0 |
| $\lambda_2$ | 0 | 0 |
| $\lambda_3$ | - | 0 |
| $\eta_{noise}$ | 1.2 | 1.1 |

Table 11: Best found hyper-parameters for the COMPROMISE environment

| HYPER-PARAMETER | MADDPG+AGENT MODELLING | MADDPG+POLICY MASK |
| --- | --- | --- |
| $\alpha_\theta$ | $1.2 * 10^{-4}$ | $2.5 * 10^{-4}$ |
| $\omega_\phi$ | 0.71 | 0.52 |
| $\tau$ | 0.0051 | 0.0077 |
| $\lambda_1$ | 0.0075 | 0 |
| $\lambda_2$ | 0 | 0 |
| $\lambda_3$ | - | 0 |
| $\eta_{noise}$ | 1.8 | 1.3 |

## E.5 Hyper-parameter search results

The performance distributions across hyper-parameters configurations for each algorithm on each task are depicted in Figure 9 using box-and-whisker plot. It can be seen that, while most algorithms can perform reasonably well with the correct configuration, TeamReg, CoachReg as well as their ablated versions boost the performance of the third quartile, suggesting an increase in the robustness across hyper-parameter compared to the baselines.

Figure 9: Hyper-parameter tuning results for all algorithms. There is one distribution per *(algorithm, environment)* pair, each one formed of 50 data-points (hyper-parameter configuration samples). Each point represents the best model performance averaged over 100 evaluation episodes and averaged over the 3 training seeds for one sampled hyper-parameters configuration. The box-plots divide in quartiles the 49 lower-performing configurations for each distribution while the score of the best-performing configuration is highlighted above the box-plots by a single dot.

# F    The effects of enforcing predictability (additional results)

The results presented in Figure 5 show that MADDPG + TeamReg is outperformed by all other algorithms when considering average return across agents. In this section we seek to further investigate this failure mode.

Importantly, COMPROMISE is the only task with a competitive component (i.e. the only one in which agents do not share their rewards). The two agents being strapped together, a good policy has both agents reach their landmark successively (e.g. by having both agents navigate towards the closest landmark). However, if one agent never reaches for its landmark, the optimal strategy for the other one becomes to drag it around and always go for its own, leading to a strong imbalance in the return cumulated by both agents. While such scenario doesn't occur for the other algorithms, we found TeamReg to often lead to cases of domination such as depicted in Figure 11.

Figure 10 depicts the performance difference between the two agents for every 150 runs of the hyperparameter search for TeamReg and the baselines, and shows that (1) TeamReg is the only algorithm that leads to large imbalances in performance between the two agents and

Figure 10: Average performance difference ($\Delta_{perf}$) between the two agents in COMPROMISE for each 150 runs of the hyper-parameter searches (left). All occurrences of abnormally high performance difference are associated with high values of $\lambda_2$ (right).

(2) that these cases where one agent becomes dominant are all associated with high values of $\lambda_2$, which drives the agents to behave in a predictable fashion to one another.

Looking back at Figure 11, while these domination dynamics tend to occur at the beginning of training, the dominated agent eventually gets exposed more and more to sparse reward gathered by being dragged (by chance) onto its own landmark, picks up the goal of the task and starts pulling in its own direction, which causes the average return over agents to drop as we see happening midway during training in Figure 5. These results suggest that using a predictability-based team-regularization in a competitive task can be harmful; quite understandably, you might not want to optimize an objective that aims at making your behavior predictable to your opponent.

Figure 11: Learning curves for TeamReg and the three baselines on COMPROMISE. We see that while both agents remain equally performant as they improve at the task for the baseline algorithms, TeamReg tends to make one agent much stronger than the other one. This domination is optimal as long as the other agent remains docile, as the dominant agent can gather much more reward than if it had to compromise. However, when the dominated agent finally picks up the task, the dominant agent that has learned a policy that does not compromise see its return dramatically go down and the mean over agents overall then remains lower than for the baselines.

# G   Analysis of sub-policy selection (additional results)

## G.1   Mask densities

We depict on Figure 12 the mask distribution of each agent for each *(seed, environment)* experiment when collected on a 100 different episodes. Firstly, in most of the experiments, agents use at least 2 different masks. Secondly, for a given experiments, agents' distributions are very similar, suggesting that they are using the same masks in the same situations and that they are therefore synchronized. Finally, agents collapse more to using only one mask on CHASE, where they also display more dissimilarity between one another. This may explain why CHASE is the only task where CoachReg does not improve performance. Indeed, on CHASE, agents do not seem synchronized nor leveraging multiple sub-policies which are the priors to coordination behind CoachReg. In brief, we observe that CoachReg is less effective in enforcing those priors to coordination of CHASE, an environment where it does not boost nor harm performance.

Figure 12: Agent's policy mask distributions. For each *(seed, environment)* we collected the masks of each agents on 100 episodes.

## G.2 Episodes rollouts with synchronous sub-policy selection

We display here and on https://sites.google.com/view/marl-coordination/ some interesting sub-policy selection strategy evolved by CoachReg agents. On Figure 13, the agents identified two different scenarios depending on the target-ball location and use the corresponding policy mask for the whole episode. Whereas on Figure 13, the agents synchronously switch between policy masks during an episode. In both cases, the whole group selects the same mask as the one that would have been suggested by the coach.

(a) BOUNCE: The ball is on the left side of the target, agents both select the purple policy mask

(b) BOUNCE: The ball is on the right side of the target, agents both select the green policy mask

Figure 13: Visualization of two different BOUNCE evaluation episodes. Note that here, the agents' colors represent their chosen policy mask. Agents have learned to synchronously identify two distinct situations and act accordingly. The coach's masks (not used at evaluation time) are displayed with the timestep at the bottom of each frame.

(a) SPREAD

(b) COMPROMISE

Figure 14: Visualization of sequences on two different environments. An agent's color represent its current policy mask. The coach's masks (not used at evaluation time) are displayed with the timestep at the bottom of each frame. Agents synchronously switch between the available policy masks.

## G.3 Mask diversity and synchronicity (ablation)

As in Subsection 7.3 we report the mean entropy of the mask distribution and the mean Hamming proximity for the ablated "MADDPG + policy mask" and compare it to the full CoachReg. With "MADDPG + policy mask" agents are not incentivized to use the same masks. Therefore, in order to assess if they synchronously change policy masks, we computed, for each agent pair, seed and environment, the Hamming proximity for every possible masks equivalence (mask 3 of agent 1 corresponds to mask 0 of agent 2, etc.) and selected the equivalence that maximised the Hamming proximity between the two sequences.

We can observe that while "MADDPG + policy mask" agents display a more diverse mask usage, their selection is less synchronized than with CoachReg. This is easily understandable as the coach will tend to reduce diversity in order to have all the agents agree on a common mask, on the other hand this agreement enables the agents to synchronize their mask selection. To this regard, it should be noted that "MADDPG + policy mask" agents are more synchronized that agents independently sampling their masks from $k$-CUD, suggesting that, even in the absence of the coach, agents tend to synchronize their mask selection.

Figure 15: (Left) Entropy of the policy mask distributions for each task and method, averaged over agents and training seeds. $H_{max,k}$ is the entropy of a $k$-CUD. (Right) Hamming Proximity between the policy mask sequence of each agent averaged across agent pairs and seeds. $\mathrm{rand}_k$ stands for agents independently sampling their masks from $k$-CUD. Error bars are SE across seeds.

# H    Scalability with the number of agents

## H.1    Complexity

In this section we discuss the increases in model complexity that our methods entail. In practice, this complexity is negligible compared to the overall complexity of the CTDE framework. To that respect, note that (1) the critics are not affected by the regularizations, so our approaches only increase complexity for the forward and backward propagation of the actor, which consists of roughly half of an agent's computational load at training time. Moreover, (2) efficient design choices significantly impact real-world scalability and performance: we implement TeamReg by adding only additional heads to the pre-existing actor model (effectively sharing most parameters for the teammates' action predictions with the agent's action selection model). CoachReg consists only of an additional linear layer per agent and a unique Coach entity for the whole team (which scales better than a critic since it only takes observations as inputs). As such, only a small number of additional parameters need to be learned relatively to the underlying base CTDE algorithm. For a TeamReg agent, the number of parameters of the actor increases linearly with the number of agents (additional heads) whereas the critic model grows quadratically (since the observation size themselves usually depend on the number of agents). In the limit of increasing the number of agents, the proportion of added parameters by TeamReg compared to the increase in parameters of the centralised critic vanishes to zero. On the SPREAD task for example, training 3 agents with TeamReg increases the number of parameters by about 1.25% (with similar computational complexity increase). With 100 agents, this increase is only of 0.48%. For CoachReg, the increase in an agent's parameter is independent of the number of agent. Finally, any additional heads in TeamReg or the Coach in CoachReg are only used during training and can be safely removed at execution time, reducing the systems computational complexity to that of the base algorithm.

## H.2    Robustness

To assess how the proposed methods scale to greater number of agents, we increase the number of agents in the SPREAD task from three to six agents. The results presented in Figure 16 show that the performance benefits provided by our methods hold when the number of agents is increased. Unsurprisingly, we also note how quickly learning becomes more challenging when the number of agents rises. Indeed, with each new agent, the coordination problem becomes more and more difficult, and that might explain why our methods that promote coordination maintain a higher degree of performance. Nonetheless, in the sparse reward setting, the complexity of the task soon becomes too difficult and none of the algorithms is able to solve it with six agents.

While these results show that our methods do not contribute to a quicker downfall when the number of agents is increased, they are not however aimed at tackling the problem of massively-multi-agent RL. Other approaches that use attention heads [14] or restrict one agent perceptual field to its $n$-closest teammates are better suited to these particular challenges and our proposed regularisation schemes could readily be adapted to these settings as well.

Figure 16: Learning curves (mean return over agents) for all algorithms on the SPREAD environment for varying number of agents. Solid lines are the mean and envelopes are the Standard Error (SE) across the 10 training seeds.