[Reviews · NeurIPS 2020]

Review 1

Summary and Contributions: The paper hypothesized and proves that coordination-promoting inductive biases on policy search helps discover successful behaviors more efficiently. Two approaches are introduced in this paper to help search for coordinated policies: TeamReg, predicts the teammate behaviors to promote coordination, CoachReg, enables agents to recognize different situations and synchronously switch to different sub-policies. Evaluation was done on continuous and discrete action control tasks. Results show that there is a significant performance improvement in almost all the domains.

Strengths: Novel idea of extending agent modeling to introduce inductive bias to promote coordination behaviours in policy search. Ability of the proposed work to run in a distributed fashion Sound evaluation proving the claims made in this paper Highly significant and relevant in MARL

Weaknesses: Would like to see a wider variety of domains for evaluation, performance on domains having higher number of agents. Results of how proposed algorithms perform when agents scale The ability of agents accurately predicting other agents actions might not be true in many domains

Correctness: Results seem to reflect the claims of this paper

Clarity: Well written paper, includes all the details necessary.

Relation to Prior Work: Prior work has been explained clearly and the paper does a good job of letting the readers know of the current state of art in the field and why and how they extended it.

Reproducibility: Yes

Additional Feedback: Why is performance of TeamReg poor, given that the soccer domain opponents are rule based (so presumably the predictability of the opponents is not necessarily a problem?)


Review 2

Summary and Contributions: Based on rebuttal and discussion: Upon reading all reviews, I recognize that we agree the article is well presented, and I stand by the concerns I raised. Note that I primarily criticized the absence of some relevant context in the original submission (which the authors admit in their rebuttal), rather than the contribution itself (albeit it may be smaller than proclaimed). The author rebuttal alleviated some concerns. Their refutation of it being a planning setting is fair. While I maintain that it is a self-play setting, this is implied by CTDE and thus not necessary to state again. A stale flavor remains from overselling their contribution’s novelty in the introduction [L36-45]. As the authors confirm in their rebuttal, it would be fair to say they extend the auxiliary task of policy prediction (established in the literature) with predictability. While the rebuttal concentrates on CTDE, my review demanded a positioning within “opponent-prediction based auxiliary tasks and intrinsic motivation”, which I see as complementary but not equivalent to CTDE. ----- This article extends the algorithm MADDPG with auxiliary tasks of (a) opponent policy prediction (this has been done before, understated or missed by the authors) and predictabilty (of the agent itself), and (b) prediction of a synchronised exploration signal used during centralised training. It argues for the need of coordinated exploration and presents empirical comparisons of the novel extended algorithms (a) and (b) against ablated versions and DDPG.

Strengths: The article addresses the critical and highly relevant challenge of joint exploration in multi-agent reinforcement learning. It is presented in good language.

Weaknesses: My concerns are mostly with the difficulty of evaluating novelty and significance given some key omissions of related work. I would expect at least a technical discussion on the difference from related work, and why this was chosen. At the very least, it should be stated that opponent policy prediction has been shown to be beneficial previously, and is here applied to MADDPG. The article also seems to carry a 'deep bias' in literature selection. The article falls below expectations in terms of related work embedding (listed below), reflection and transfer insights. The specific algorithmic extensions seem motivated a posteriori, but the method of extending is not easily transferable. The motivation section focusses on action re-encoding, but does not reflect on this, leaving a perceived gap towards the extensions. Results are not clear cut, and require a lot of exceptions in Section 7.1.

Correctness: Overall, the article's technical arguments seem correct. The claim of novelty is questionable given the omissions of some key related work, and the lack of technical juxtaposition with some of what is cited (see related work comments below).

Clarity: The presentation is good, both in language and layout. The arguments that are made are easy to follow, albeit the related work section comes a bit late and remains superficial. The abstract does not explicitly state that this article addresses the planning setting (having access to the model), and operates in self-play (controlling all learning agents, especially for CoachReg).

Relation to Prior Work: - One key omission is the reference "A Deep Policy Inference Q-Network for Multi-Agent Systems" by Hong et al, 2018, which employed a KL-divergence opponent policy prediction loss (compare Eq. (4), (5) of both papers). Also, while [11] is cited, their use of cross-entropy opponent policy prediction is neither mentioned in Section 4.1 nor 5. These two omissions put a questionmark to the purported novelty and originality of the first extensions (called (a) in the summary above). - The introduction should highlight the line of work this article falls into, which to me is opponent-prediction based auxiliary tasks and intrinsic motivation. Some links are drawn in related work, but I would expect the high level positioning in the introduction. - The algorithm "MAVEN: Multi-agent variational exploration", co-authored by Shimon Whiteson, deserve mentioning and demarcation, as it similarly addresses the fundamental challenge of joint exploration. - The link of predictability to intrinsic motivation is recognized but not fully pursued or exposed to the reader. - The policy mask is not embedded in previous work, while it has similarities with attention models and hierarchical models (e.g., "Feudal networks for hierarchical reinforcement learning").

Reproducibility: Yes

Additional Feedback: - It is unclear (from the main article) if negative regularization parameters lambda are searched. - KL divergence is first abbreviated and later written in full. - "MADDPG, the most widely used algorithm" is an unsubstantiated overstatement - "This through evaluation" maybe 'thorough'? - While the broader impact section is fine, it could be strengthened by highlighting that joint exploration is indeed a bottleneck challenge in MARL.


Review 3

Summary and Contributions: The paper proposes two novel techniques to encourage the discovery of coordinated strategies in the centralized training decentralized execution (CTDE) framework.

Strengths: The paper is excellent in clarity. The paper clearly presents and explains the proposed techniques. The details of the experiments are given to make the work highly reproducible. The proposed techniques are compatible with any MARL algorithm in the Centralized Training and a Decentralized Execution (CTDE) framework, and thus the paper might be interesting to the broad research community in MARL.

Weaknesses: The empirical evaluation would be much more convincing if the proposed algorithms are compared with something stronger than MADDPG (and its variants). To your knowledge, is there any other work that aims to improve the centralized training in the CTDE framework? The introduction of the two techniques are somewhat ad-hoc, even after section 3 is presented for motivation. In particular, the ending of section 3 emphasizes on learning “the same type of constraint”, while the TeamReg seems to focus on purely predicting others’ decisions. Moreover, in section 4.2, the idea of policy masks is introduced without enough motivation/intuition. Is the idea of policy masking itself novel?

Correctness: Yes.

Clarity: Yes, the paper is well written in terms of clarity.

Relation to Prior Work: Yes.

Reproducibility: Yes

Additional Feedback: Given that both TeamReg and CoachReg reduce to auxiliary losses, is there any reason why the authors didn't combine these losses? --- I am lowering my score a bit, but I am still learning towards accepting the paper.


Review 4

Summary and Contributions: This paper addresses how to learn collective behavior across multiple agents without succumbing to the curse of dimensionality and without resorting to single agent exploration. The key idea in the paper is that by coordinating the agents’ policies in order to guide their exploration it is possible to train multi-agent systems to exhibit competitive coordinated behavior. This policy coordination (or regularization) is a form of inductive bias. The paper presents two methods: TeamReg, based on inter-agent action predictability and CoachReg that relies on synchronized behavior selection. Empirical evaluation is performed on continuous control tasks with sparse rewards and on the discrete action Google Research Football environment.

Strengths: A key strength of the paper are two methods for policy coordination in a multi-agent team. As the authors acknowledge, their focus is on agent behaviors that are predictable given teammate behavior. This is a reasonable basis on which to build. The second strength of the paper is to point out that the widespread assumption that centralized training always outperforms decentralized training may not be valid.

Weaknesses: The key limitation of the work is that the empirical evaluation is inconclusive. The results in section 7.1 illustrate that on the COMPROMISE and CHASE environments more work remains to be done. On the BOUNCE environment more episodes are needed to clarify asymptotic behavior. The results in section 7.4 are more promising (how did you decide that 80K episodes was a reasonable number?) though I'm puzzled why MADDPG + TeamReg does not perform somewhat better. While MADDPG + CoachReg does seem to learn policies that achieve positive return the numbers overall are still small and it's not clear whether these results imply anything for scaling up to scenarios more complicated than 3v2.

Correctness: The empirical results are a promising start but needs more work to be conclusive.

Clarity: The paper is well-written and easy to read and understand.

Relation to Prior Work: The paper clearly makes a connection to prior work in the field and how it differs from previous contributions. It is well-situated in the literature.

Reproducibility: Yes

Additional Feedback: The extension of the baseline training (inlined figure in rebuttal) is appreciated.

[Author Response · NeurIPS 2020]

We would like to thank the reviewers for their thorough evaluations and for bringing to our attention some missing citations and typos. We answer specific questions raised by the reviewers, below.

***Performance w.r.t. number of agents (R1).*** We discuss how our methods scale with the number of agents in Appendix H. Specifically, Figure 16 shows that our method's benefits hold but that the underlying algorithm (MADDPG in this case) fails to handle many agents; this has also been shown in [12].

***Related work and novelty (R2, R3).*** *(To R2 and R3)* We are grateful for bringing to our attention some relevant work in hierarchical RL. Importantly, however, the novelty of CoachReg does not lie in training sub-policies (which are obtained here through a simple and novel masking procedure) but rather in co-evolving synchronized sub-policies across multiple agents. This is indeed closer to the joint exploration work of Mahajan et al. (2019)[1] (as pointed out by R2). Yet, a major difference is that MAVEN's situational-prediction occurs only on the first timestep and requires synchronized random seeds across the agents *at test time*, whereas with CoachReg agents explicitly learn a set of subpolicies, of which they choose one to execute at every timestep. Each agent chooses without using a common sampling procedure and execution is therefore fully decentralized. We will update our manuscript with these more explicit clarifications. *(To R2)* Thank you for referring us to Hong et al.[2] whose method is very close to "MADDPG + agent-modelling", the TeamReg ablation that we compare against in Section 7, Figure 5. As we discuss in Related Work (L211-214), agent-modelling (through cross-entropy prediction) is now a widely used MARL component and Hong et al., like [11], uses it as an auxiliary task to learn richer representations. Similarly, TeamReg relies on agent-modelling (L141-146) but our contribution with TeamReg is to instead use it to explicitly influence other agents' behavior toward being predictable rather than just learning a representation (L145-148 and L219-222). To our knowledge, this is a novel contribution and has not been considered in prior work.

***Positioning of the paper and missing keywords (R2).*** While the high level positioning of this work in the Centralized Training Decentralized Execution framework (CTDE) is already made clear throughout the paper (L20, 38, 42, 237, 266, 323), we will highlight it in the abstract as suggested. However, we do not believe that "the planning setting" (usually referring to making use of a transition model rather than the agents model) or "self-play" (where an agent, short of having an opponent to train with, plays against itself) are relevant keywords for our work.

***Importance to the broader community, reflection, motivation and transfer (R2).*** As highlighted by other reviewers, our work makes significant contributions to the research community: at a high level we question the widespread assumption that centralized training always outperforms decentralized training, proposing a definition for coordinated behavior (based on behavior predictability), in order to improve upon it. We propose two (2) novel practical coordination promoting methods that are applicable to any CTDE algorithm and evaluate them on three (3) different baselines based on the prevalent MADDPG algorithm, as well as two (2) ablated versions of our methods.

***MADDPG baseline (R3).*** We disagree with the premise that MADDPG is a weak baseline and argue that the evaluation setting plays a major role in allowing valid and insightful experimental results. Several recent works have pointed out that hyperparameter tuning often plays a fundamental role in determining which algorithms best perform at a given task (Henderson et al. (2018)[3], Colas et al. (2019)[4]). In our work, we make a substantial effort to offer fair and significant comparisons by allowing our three (3) baselines (DDPG, MADDPG, MADDPG + sharing) and two (2) ablations (MADDPG + agent modelling, MADDPG + policy masks) a full hyperparameter tuning, yielding a competitive suite of baselines. To substantiate the importance of such re-tuning, we provide here additional experiments reporting the improvements of our tuned MADDPG over MADDPG with the original hyperparameters configuration from [22]. The improvements are 900 % (SPREAD), 1300 % (BOUNCE), 700 % (COMPROMISE) and 400 % (CHASE) and highlight the important performance gains allowed by our evaluation procedure to the baselines.

***Conclusiveness of the results (R4).*** As requested, we extended the training of the baselines and the inlined figure shows that our methods still outperform them. Additionally, we believe that our evaluation is sound, conclusive and substantiates our claims (as concluded by R1). A key question here is "do the proposed coordination-inducing methods improve performance of the CTDE framework?". We answer this by examining the impact of our proposed

ideas on the widely used MADDPG CTDE algorithm and we perform an ablation study to probe each element of our contribution in more detail. We apply a careful experimental methodology (Tables 2 through 11) to both continuous and discrete action environments of varying complexity, requiring *significant* computing resources: e.g., the retraining in Table 1 of our submission alone require 120 CPU-days and prevented us from extending the number of training steps illustrated in this rebuttal.

[1] Mahajan et al., MAVEN : Multi-Agent Variational Exploration (2019)  [2] Hong et al. A Deep Policy Inference... (2018)
[3] Henderson et. al., Deep RL that matters. (2018)  [4] Colas et al., A Hitchhiker's Guide to Statistical Comparisons of RL ... (2019)


[Meta-Review · NeurIPS 2020]

Originally, there was some disagreement between reviewers on this paper, but after rebuttal and careful discussion between reviewers and AC, all agree that the paper is interesting and has merit and could be proposed for acceptance as poster. One critical reviewer now recognises that the predictability idea is neat and the concern about positioning of the work has been largely clarified. Reviewers agree there is a contribution to joint exploration in MAS, which is one of the bottlenecks that deserve being addressed and discussed.